# Cyclophosphamide Induces Lipid and Metabolite Perturbation in Amniotic Fluid during Rat Embryonic Development

**DOI:** 10.3390/metabo12111105

**Published:** 2022-11-12

**Authors:** Jianya Xu, Huafeng Fang, Ying Chong, Lili Lin, Tong Xie, Jianjian Ji, Cunsi Shen, Chen Shi, Jinjun Shan

**Affiliations:** 1Jiangsu Key Laboratory of Pediatric Respiratory Disease, Nanjing University of Chinese Medicine, Nanjing 210023, China; 2School of Medicine & Holistic Integrative Medicine, Nanjing University of Chinese Medicine, Nanjing 210023, China; 3Medical Metabolomics Center, Nanjing University of Chinese Medicine, Nanjing 210023, China

**Keywords:** cyclophosphamide, amniotic fluid, lipidomics, embryotoxicity, oxidative stress

## Abstract

Cyclophosphamide (CP) has been proven to be an embryo-fetal toxic. However, the mechanism responsible for the toxicity of the teratogenic agent has not been fully explored. This study aimed to examine the teratogenicity of CP when administered in the sensitive period of pregnant rats. The effect of CP on the lipid and metabolic profiles of amniotic fluid was evaluated using a UHPLC-Q-Exactive Orbitrap MS-based method. Metabolome analysis was performed using the MS-DIAL software with LipidBlast and NIST. Initially, we identified 636 and 154 lipid compounds in the positive and negative ion modes and 118 metabolites for differential analysis. Mainly 4 types of oxidized lipids in the amniotic fluid were found to accumulate most significantly after CP treatment, including very-long-chain unsaturated fatty acids (VLCUFAs), polyunsaturated fatty acid (PUFA)-containing triglycerides (TGs), oxidized phosphatidylcholine (PC), and sphingomyelin (SM). Tryptophan and some long-chain saturated fatty acids were lowered pronouncedly after CP treatment. These findings suggest that CP may exert teratogenic toxicity on pregnant rats through maternal and fetal oxidative stress. The UHPLC-Q-Exactive Orbitrap MS-based lipidomics approach is worthy of wider application for evaluating the potential toxicity of other agents (toxicants) during embryonic development.

## 1. Introduction

Cyclophosphamide (CP) is a common chemotherapy agent and immunosuppressive agent activated in the liver. However, it is of concern for several reasons, including the strong mutagenesis and teratogenesis of its metabolites [1]. CP has been shown to cause embryo or fetal resorption, fetal growth restriction in body weight and length, and fetal malformations, particularly organ and skeletal malformations [2,3,4,5]. CP-induced oxidative stress has been considered a primary mechanism responsible for teratogenesis [6]. However, the effect of CP-induced developmental toxicity has not been precisely interpreted.

Metabonomics (or metabolomics) measures the metabolic profiles of a wide variety of biological samples and enables complicated qualitative and quantitative analysis of endogenous small molecule metabolites [7]. It has been widely used to elucidate detailed mechanisms in toxicology research [8], such as identification of toxic biomarkers by screening metabolites with altered concentrations after treatment and metabolic pathways related to injury and poisoning. Metabonomics cannot be confined to the blood and urine analysis but is more advisable for amniotic fluid analysis to track embryo development within the amniotic sac due to the non-invasive sampling [9]. 

Amniotic fluid comprises fetal urine and the mother’s oral, nasopharyngeal, tracheal secretions, and pulmonary fluids transferred into the amniotic sac across the amniotic membrane [10]. It serves as a reservoir critical for embryo development and fetal survival and growth during pregnancy, when the composition of amniotic fluid constantly changes [11,12], which can be directly influenced by the fetus, particularly metabolites from the placenta, fetal skin, lungs, and gastric juice over time [13]. Amniotic fluid is rich in lipids, which are predominant in pulmonary surfactants. Lipids can be exchanged between the fetal lung and amniotic fluid through swallowing and alveolar lavage physiologically. Therefore, some metabolites of the complex, dynamically changing body fluid can serve as biomarkers to reflect the physiological or pathological state of the developing fetus when exposed to exogenous factors such as drugs. Amniotic fluid metabonomics can accurately examine the effects of drugs and toxicants on the fetus and identify biomarkers closely associated with the metabolic profile of amniotic fluid in the pathological state [14]. 

Studies on amniotic fluid metabonomics, particularly metabonomics of amniotic fluid lipids, after CP treatment have not been reported so far. In this study, we examined metabonome and lipidome changes in the amniotic fluid after CP administration using gas chromatography–mass spectrometry (GC-MS) and liquid chromatography–mass spectrometry (LC-MS). This study aimed to investigate the mechanism for CP-induced developmental toxicity in rat embryos. Our study is worth referencing for identifying other drugs (toxicants) with potential developmental toxicity.

## 2. Materials and Methods

### 2.1. Chemicals and Reagents

Cyclophosphamide for injection (batch number 9B285A) from Baxter Oncology GmbH, Halle, Germany; internal standard, lyso PE (17:1) (batch number: LM171LPE-11), SM (17:0) (batch number: 170SM-13), PE (17:0/17:0) (batch number: LM170PE-19) from Avanti Polar Lipids, Alabaster, AL, USA; methyl tert-butyl ether (MTBE), N-hexane, isopropanol, ammonium acetate and ammonium formate (99.8% mass spectrometry pure) from ROE Scientific, Newark, NJ, USA; glacial acetic acid (batch number 15080553613, purity: 99.5%) from Nanjing Chemical Reagent Co., Ltd., Nanjing, China; acetonitrile and methanol (99.8% mass spectrometry pure) from Merck, Darmstadt, Germany; methoxyamine hydrochloride, pyridine, N, O-Bis (trimethylsilyl) triflfluoroacetamide (BSTFA, batch number 15238) with 1% trimethylchlorosilane (TMCS), formic acid and 1,2-13C-myristic acid from Sigma Aldrich, St. Louis, MO, USA; methanol from Supelco, Bellefonte, PA, USA; and malonaldehyde (MDA) and superoxide dismutase (SOD) activity detection kit from Wuhan Abbkine Company, Wuhan, China. 

### 2.2. Embryo-Fetal Toxicity

#### 2.2.1. Animals

Healthy 11-week Sprague–Dawley (SD) rats were obtained from Vital River Laboratory Animal Technology Co., Ltd. (Beijing, China; 1100111911042043). All experimental procedures were approved by the Institutional Animal Care and Use Committee of Nanjing University of Chinese Medicine (201906A037) and conducted according to the committee guidelines.

All animals were housed at a constant temperature of 23–25 °C, relative humidity of 55%, and 12 h light/12 h dark cycle (08:00–20:00). They were given free access to water and common foodstuffs to acclimatize before experiments. Mating was carried out at 4 p.m. by putting two females and one male together. The males were removed at 8 a.m. the next morning. The vaginal smears of female rats were collected and observed under a microscope. The day we initially observed sperms in the samples was designated as gestation day 0 (GD0). We repeated the mating process until enough mated female rats were obtained.

#### 2.2.2. Treatment and Sample Collection

Pregnant females were randomly divided into the CP (9B285A, Anderson & Baxter, Plymouth, MN, USA; subcutaneous injection of 15 mg·kg^−1^ cyclophosphamide) and CK (isovolumetric normal saline) groups on GD12, with ten in each group. Animals were anesthetized and sacrificed on GD17 after being fasted for 10 h to collect organ samples. The heart, liver, kidney, and part of the placentas were fixed with 4% paraformaldehyde for histopathologic evaluation.

Uteri with contents were removed and weighted. Amniotic fluid samples were drawn from each amniotic cavity with a syringe needle, and all the samples of one pregnant rat were mixed together and immediately stored at −80 °C. Some placentas were also isolated and stored at −80 °C. The resorbed, viable, and dead fetuses were counted, and the weights and tail lengths of the fetuses were recorded.

#### 2.2.3. Oxidative Stress Assessment

The fixed placenta and other tissue samples were embedded and sliced. The sections were stained with hematoxylin and eosin and observed under a light microscope for histopathological scoring [15]. A colorimetric method was utilized to determine the oxidative stress indices MDA and SOD in the placenta. In brief, 1 g of placental tissue was weighed and homogenized in ice-cooled lysis buffer. The mixture was centrifuged at 13,000× *g* for 10 min, and the supernatant was collected for analysis. The samples were mixed with working reagents using MDA concentration and SOD activity detection kits (Wuhan Abbkine, Wuhan, China) and incubated according to the manufacturer’s instructions. The absorbance of SOD was measured at 450 and that of MDA at 532 and 600 nm in a microplate reader. The results expressed as MDA content and SOD activity were calculated according to the manufacturer’s specifications.

#### 2.2.4. Statistical Analysis

GraphPad Prism 9.1.0 (GraphPad Software, San Diego, CA, USA) was used for data analysis. Differences between the two groups were compared using the Mann–Whitney non-parametric test. Data are expressed as mean ± standard deviation (SD). A *p*-value of <0.05 was considered statistically significant.

### 2.3. Untargeted Lipidomics and Metabolomic Analysis

#### 2.3.1. Sample Preparation

Lipid and metabolite extraction from amniotic fluid samples was performed using liquid–liquid extraction with methyl tert-butyl ether (MTBE; ROE, Newark, NJ, USA) before analysis [15,16,17,18]. In brief, 500 μL of amniotic fluid was lyophilized and redissolved in 200 μL of deionized water. Then, 80 μL of redissolved amniotic fluid was pipetted off into a 1.5 mL centrifuge tube containing 225 μL of ice-cooled methanol (Merck, Darmstadt, Germany) pre-mixed with lyso PE (17:1; LM171LPE-11), SM (17:0; 170SM-13), and PE (17:0/17:0; LM170PE-19) internal standards (5 μg·mL^−1^; Avanti Polar Lipids, Alabaster, AL, USA). The solution was vortexed for 10 s, and 750 μL of ice-cooled MTBE was. The mixture was shaken for 10 min at 4 °C, supplemented with 188 μL of deionized water, vortexed for 20 s, and centrifuged at 14,000 rpm for 2 min at 4 °C. In all, 350 μL of the upper layer (the organic phase, mainly including lipids) and 110 μL of the bottom layer (the aqueous phase, mainly including polar substances) were separately transferred to a new centrifuge tube (1.5 mL). The samples were dried using the Savant SPD1010 vacuum centrifugal concentrator (Thermo Fisher Scientific, Waltham, MA, USA) and stored at −20 °C before testing. Lipids in the upper layers were lysed with 110 μL of methanol–toluene (9:1) solution. Liquid chromatography–mass spectrometry (LC-MS) analysis was performed using the Q-Exactive Quadrupole-Electrostatic Field Orbitrap High-Resolution Mass Spectrometer (Thermo Fisher Scientific, Waltham, MA, USA). The dried substances (the polar fraction) from the bottom layer were supplemented with 2.5 μg of 1,2-C^13^ myristic acid (diluted with pyridine; Sigma Aldrich, St. Louis, MO, USA) and 30 μL of pyridine solution with 10 mg·mL^−1^ methoxyamine hydrochloride (Sigma Aldrich, St. Louis, MO, USA). They were mixed for 5 min and shaken at 300 r·min^−1^ for 1.5 h at 30 °C. Then, 30 μL of N, O-Bis (trimethylsilyl) triflfluoroacetamide (BSTFA; 15238, Sigma Aldrich, USA) was added to them in a laboratory fume hood, and they were mixed for 1 min and shaken at 300 r·min^−1^ for 0.5 h at 37 °C. Finally, the samples were centrifuged at 18,000 r·min^−1^ for 10 min, and 50 μL of the supernatant was used for gas chromatograms and mass spectrometry (GC-MS) analysis. A total of 20 μL of amniotic fluid was taken from each sample to prepare the quality control (QC) sample using the procedures described above. QC samples were injected every 10 samples to assess the reliability and stability of the system.

#### 2.3.2. Chromatographic Separation and Mass Spectrometer Settings

In LC-MS lipidomics, 2 µL aliquots of the amniotic fluid samples were injected on a Waters Acquity UPLC CSH C18 column (100 mm × 2.1 mm, 1.7 m), employing water and acetonitrile (6:4; Merck, Germany) as the gradient mobile phase A and isopropanol (ROE, USA) and acetonitrile (9:1, both containing 10 mM ammonium formate and 0.1% formic acid) as the gradient mobile phase B, at a flow rate of 0.3 mL·min^−1^ and a constant temperature of 60 °C [15]. The elution procedure was as follows: 0–4.0 min, 15% B; 4.0–5.0 min, 15–48% B; 5.0–22.0 min, 48–82% B; 22.0–23.0 min, 82–99% B; 23.0–24.0 min, 99% B; 24.0–24.2 min, 99–15% B; 24.2–30.0 min, 15% B. The spray voltages were 3.5 kV and 3.0 kV for operation in the positive and negative ion modes. Other settings for operation in both ion modes include sheath gas pressure 35 arbitrary unit (Arb), aux gas pressure 15 Arb, capillary temperature 325 °C, and heater temperature 300 °C. 

Temperature-programmed GC-MS analysis was performed for the polar fraction of the amniotic fluid samples (described in 2.3.1) using the method described elsewhere [15]. Briefly, 1 μL aliquots of derived amniotic fluid samples were injected on a TG-5MS capillary column (30 m × 0.25 mm, 0.25 μm) with helium as a carrier gas (flow rate 1.2 mL·min^−1^, split ratio 20:1, and injector temperature 250 °C). GC oven temperature was programmed as follows: 60 °C for 1 min, 20 °C/min to 320 °C, and 320 °C for 5 min. The system was equipped with an electron ionization (EI) ion source and was operated according to the following settings: ion source temperature 280 °C, transfer line temperature 250 °C, ionization energy 70 eV, acquisition time 3.5–19.0 min, and MS scanning range *m*/*z* 50–500.

#### 2.3.3. Data Processing and Statistical Analysis

LC-MS and GC-MS metabolomics data were processed and analyzed using MS-DIAL (MS-DIAL software, version: 3.0.0.0, Japan). Peaks and components were identified according to the NIST database mass spectra for lipids and related metabolites. Their data matrices, including name, peak height, and classification, were subjected to MetaboAnalyst 5.0 (metaboanalyst.ca/faces/ModuleView.xhtml, accessed on 21 June 2022) for multivariate analyses, heatmap analysis, clustering, and visualization to identify principal components. We utilized the *U* test and FDR calibration to obtain differential metabolites between pregnant rats receiving CP and normal saline (*p* < 0.05, FDR < 0.2, and FC > 1.5). Further, we performed ChemRICH enrichment analysis (http://chemrich.fifiehnlab.ucdavis.edu/, accessed on 28 July 2022) of differential lipids to characterize their expressions after CP treatment [19].

## 3. Results

### 3.1. CP disturbs Embryonic Development

Pregnant rats treated with CP were markedly emaciated and lost weight (Table 1) versus those treated with normal saline (the CK group, or CK). Three rats had miscarriages after CP treatment versus one in CK. Anatomical and histopathological observations showed no obvious pathological injury in the heart, liver, and kidney in pregnant rats in CP versus CK. Blood clots were found in the placental vessels of two rats whose fetuses either were resorbed or died during late gestation after CP treatment (Figure 1). Rats treated with CP showed a reduced volume of amniotic fluid, which was dark brown, compared to those treated with normal saline.

As shown in Table 1 and Figure 2, rats treated with CP showed markedly lower fetal and placental weights (*p* < 0.05), a higher percentage of late fetal death (*p* < 0.05), and significant reductions in fetal weight and length (*p* < 0.01) on gestation day (GD) 17.

### 3.2. Lipidomic Profiling of Amniotic Fluid after CP Treatment

#### 3.2.1. Lipid Components in the Amniotic Fluid

Lipid components in CK and CP were identified using UHPLC-Q-Exactive Orbitrap MS and analyzed using MS-DIAL with the LipidBlast library that stores MS/MS spectra of over 200,000 lipids. Amniotic fluid samples were randomly detected in positive and negative ion modes. The total ion current (TIC) showed lipid distribution in positive and negative ion modes (Appendix A) during the retention time in MS spectrometry. Mass accuracy and MS/MS matching were performed with LipidBlast to annotate fragment ions. The primary and secondary fragment ions (Appendix A) were matched to LipidBlast to identify lipid compounds and the corresponding strains.

Finally, 636 lipid compounds were identified in the positive ion mode, primarily including cardiolipin (CAR), cholesteryl ester (CE), ceramides (CER), diacylglycerol (DG); lysophosphatidylcholine (lysoPC), lysophosphatidylethanolamine (lysoPE), phosphatidylcholine (PC) phosphatidylethanolamine (PE), phosphatidylinositol (PI), phosphatidylserine (PS), sphingomyelin (SM), and triglycerides such as triglyceride (TG). In addition, 154 lipid compounds in the negative ion mode were confirmed. The top eight lipid strains were as follows: fatty acid (FA), lysophosphatidylcholine (lysoPC), lysophosphatidylethanolamine (lysoPE), phosphatidylcholine (PC), phosphatidylethanolamine (PE), phosphatidylinositol (PI), phosphatidylserine (PS), and sphingomyelin (SM).

#### 3.2.2. UHPLC-Q-Exactive Orbitrap MS Method Validation

For improved UHPLC-Q-Exactive Orbitrap MS resolution, three methods were employed to calculate the experimental error as described elsewhere [20]. Sample normalization methods were utilized to calculate the relative standard deviations (RSD) values of QC samples to assess the stability and reproducibility of the LC-MS system. The smallest RSD (10.04%) in the positive ion mode was obtained after PQN normalization (Appendix A). The smallest RSD (7.011%) in the negative ion mode was observed after linear normalization (Appendix A). The results indicate that the LC-MS system was stable throughout the analysis. 

#### 3.2.3. Non-Targeted Lipidomics Analysis

Differences in amniotic fluid lipid metabolites between CP and CK were compared in the positive and negative ion modes, and data sets were analyzed using unbiased PCA (Figure 3) and OPLS-DA (Figure 4). Permutation tests of OPLS-DA models showed that each model had a low risk of overfitting in the positive and negative ion modes (*p* < 0.01). OPLS-DA analysis revealed satisfactory accuracy (R2Y = 0.998 in the positive ion mode and R2Y = 0.998 in the negative ion mode) and predictive ability of models (Q2 = 0.709 in the positive ion mode and Q2 = 0.785 in the negative ion mode) (Figure 4A–D). The results showed a significant difference in lipid component alterations between CP and CK in the positive and negative ion modes. This indicates that CP can result in metabolic alterations in the amniotic fluid in pregnant rats.

#### 3.2.4. CP Induces Lipid Perturbation in the Amniotic Fluid

Amniotic fluid lipid metabolites differentially expressed between CP and CK were identified using FDR-corrected *U*-test (*p* < 0.05, FDR < 0.2, and fold change (FC) > 1.2). Those with FC > 1.5 were considered significantly associated with CP treatment. Initially, 45 and 22 lipid compounds in the positive and negative ion modes were recognized as differentially expressed in CP versus CK (Appendix A). Among them, 12 lipids such as TG and lysoPC (Figure 5A) and 21 lipids such as FA, SM, PI, and PC (Figure 5B) in the positive and negative modes were closely associated with PC treatment.

According to concentrations determined by fragment ion peaks in MS2, the top 25 differential lipids were summarized in the heatmaps (Figure 6A,B). Lipid distribution was obviously different in CP versus CK. To clarify the differences, differential lipids in the positive and negative ion modes were subjected to ChemRICH enrichment analysis for clustering and the Kolmogorov–Smirnov test to determine the significance of the cluster. The lipid cluster with a *p* < 0.05 was considered significantly affected by CP treatment. We found that the majority of affected lipids were TGs and FAs, with their concentrations changed most after CP treatment (Figure 6C). Specifically, concentrations of most unsaturated TGs and all unsaturated Fas were upregulated after CP treatment, and those of all saturated TGs were downregulated. This opposite expression trend was also found in unsaturated and saturated lysoPCs. Parts of PIs, oxidative PCs, and oxidative SMs were elevated after CP administration (Figure 6D).

As for differential lipid metabolites, CP stimulated FAs, SMs, PCs, PIs, and some TGs and lowered the levels of some TGs, LPC, and Cer in the amniotic fluid compared to normal saline treatment (Figure 5 and Figure 6). These changes in the amniotic fluid lipid landscape indicate lipid perturbation in pregnant rats after CP intervention.

### 3.3. Metabolic Profiling of Amniotic Fluid

Amniotic fluid metabonomics was performed with the GC-MS system, as described in Section 2.3.2. GC-MS TIC of QC samples showed clear peaks, indicating that the system was able to quantify metabolites (Appendix A). QC samples were clustered after PQN normalization (Appendix A), suggesting that the system was stable throughout the analysis. 

Ultimately, 118 metabolites were identified. These data were input into Metaboanalyst 4.0 for unbiased PCA and OPLS-DA analysis to identify differential metabolites in CP versus CK. OPLS-DA score plots implied changes in the metabolite landscape or altered concentrations of some metabolites after CP treatment (Figure 7A). A total of 1000 permutation tests showed that OPLS-DA models were reliable (R2Y = 0.98 and Q2 = 0.63; Figure 7B) for predicting variables responsible for clustering, with a certain risk of overfitting. Then, data were log-transformed and subjected to one-way ANOVA for the screening of differential metabolites (*p* < 0.05, FDR < 0.2, and FC > 1.5). Four amniotic fluid metabolites were identified. CP significantly suppressed stearic acid (SA), palmitic acid (PA), and tryptophan levels and elevated the glycerol-1-phosphate level (Figure 7D, Appendix A). 

### 3.4. CP Triggers Placental Oxidative Stress

As shown in Figure 8, CP pronouncedly stimulated the malonaldehyde (MDA) concentration in the placenta of pregnant rats compared to those receiving normal saline (*p* < 0.05). However, superoxide dismutase (SOD) activity was markedly inhibited after CP treatment (*p* < 0.01).

## 4. Discussion

CP is an effective antitumor agent with multiorgan toxicity, particularly pulmonary toxicity, in addition to the known hepatorenal toxicity [21,22]. For pregnant women, CP may induce severe developmental toxicity (e.g., embryo malformations, embryo resorption, stillbirth, and growth retardation) by passing through the placental barrier directly or permeating into the microenvironments within and outside the uterus indirectly. Kim, S.H. et al. [3,6] reported emaciated appearance and weight loss caused by intra-pregnancy CP administration, consistent with our findings (Table 1).

As amniotic fluid is the fluid that surrounds the embryo during gestation, several components have been identified as indicators for monitoring fetal and maternal health. Lipids are among the most important components of the amniotic fluid, primarily derived from fetal alveolar fluid and skin exudate. Better knowledge of the amniotic fluid lipid composition can help to explore the mechanism of how toxic agents disturb embryonic development. Lipidomics is a sophisticated tool for identifying potential lipid biomarkers in diseases and their roles in bioactivities (such as development) by comparing physiological and pathological lipid landscapes [23,24]. In this study, we employed a UHPLC-Q-Exactive Orbitrap MS-based lipidomics approach to explore lipid metabolism in the amniotic fluid after CP treatment. Initially, 890 lipid compounds were identified (e.g., FA, lysoPC, lysoPE, PC, PE, PI, PS, TG, and SM), including 636 in the positive ion mode and 154 in the negative ion mode. Among them, we screened differential lipids closely associated with CP, including FAs (mostly unsaturated FA), SMs, PCs, PIs, and the majority of TG (mostly unsaturated TG) which were significantly upregulated and a small number of TGs (particularly saturated TG), LPC, and Cer, which were significantly downregulated (Figure 5).

### 4.1. FAs

In this study, CP was found to markedly increase the levels of amniotic fluid fatty acids (FAs), particularly very-long-chain fatty acids (VLCFAs) with carbon chain length over 20 (e.g., FA 22:4, FA 24:4, and FA 26:5; Figure 5). VLCFAs are critical for the formation of the skin barrier and the phospholipid molecular layer of cells and responsible for several biofunctions, such as nutrient storage, cell-to-cell communication, and cellular biochemical reactions [25,26]. However, the hydrophobic properties of FAs determine that excessive VLCFAs may exert cytotoxic effects to induce cell damage, such as oxidative stress and inflammation [27,28]. Laura R. Parisi, L.R. et al. reported that excessive ultra-long-chain fatty acids are essential in triggering multiple mechanisms of membrane destruction, thus facilitating cell death during necrotizing apoptosis [29,30]. VLCFA accumulation in the plasma and skin-derived fibroblasts of X-linked adrenoleukodystrophy patients has been shown to have significant associations with fatal neurodegenerative phenotypes, including adrenal medulla neuropathy (AMN) and childhood-onset cerebral adrenoleukodystrophy (CCALD) [31]. Therefore, our finding of elevated amniotic fluid VLCFAs after CP treatment indicates that CP may exert toxic effects on embryonic development by disturbing VLCFA metabolism.

### 4.2. SMs

SMs are the most important sphingolipids that constitute the cell membrane and can be categorized into long (C14–C18), medium-long (C18–C20), very-long (C18–C24), and ultra-long (>C24) SMs based on the carbon number in the fatty acyl chain [32]. SM metabolites such as ceramide (Cer), sphingosine (Sph), and sphingosine-1-phosphate (S1P) are involved in multiple bioactive signalings associated with cellular functions, such as cell cycle, cell death, cell growth, cell senescence, autophagy, metabolism, inflammation, stress response, and immune response [33]. Cytotoxic DNA-damaging CP, among others, has been reported to significantly alter sphingomyelin (SM) levels [34]. Consistently, our results showed that several amniotic fluid SMs, particularly oxidized ones such as SM 40:3;2O, were markedly elevated after CP treatment (Figure 5 and Figure 6D and Appendix A).

Intracellular SM hydrolysis is catalyzed by sphingomyelinases (SMases) in lysosomes and generates ceramide and phosphocholine. Congenital defects in phospholipase—often leading to Niemann–Pick disease—result in SM accumulation in tissues, leading to enlarged liver and spleen or severely impairing the central nervous system, which can be life-threatening [35]. Thus, SMs may serve as a driving force of neural development and neurodegeneration [36,37]. SMs are critical for supporting skin barrier functions [38]. The loss of functions of several enzymes involved in SM metabolism will lead to the deficit of specialized long-chain ceramides in the skin, thus being unable to form an impermeable barrier [39]. In this study, a lower level of long-chain ceramide was detected in the amniotic fluid in pregnant rats receiving CP versus normal saline (Appendix A). This indicates that CP may impair the development of the skin barrier function and embryonic nervous system by disrupting SM metabolism.

### 4.3. PIs

Inositols are involved in lipid synthesis, cell membrane formation, cell growth and differentiation, and the maintenance of cell morphology [40]. Intracellular inositol derivatives can be roughly classified into phosphatidylinositol (PI), phosphatidylinositol 4-phosphate (PIP), and 4,5-diphosphate (PIP2) [41], with each responsible for various functions by activating corresponding metabolic pathways. High inositol concentrations detectable in the myocardium, skeletal muscle, and central nervous system in embryos indicate an increased demand for this nutrient during embryonic development [42]. During embryogenesis, inositols serve as an osmolyte to enlarge amniotic and coelomic cavities, precursors for cell membrane formation, and substrates of the pentose phosphate pathway essential for nucleic acid synthesis [43]. Inositol supplementation in perinatal women has been shown to reduce the risk of recurrent neural tube defects (NTD) in fetuses [44]. Overall, fetal uptake of inositol is essential for embryonic development.

Higher inositol concentrations in the embryonic compartment than in the maternal serum are detectable in the early weeks of human gestation (5–12 weeks) [43]. This indicates that active, carrier-mediated inositol transport is present during early gestation. Osmotic inositols can be absorbed in embryos through the sodium–myo-inositol transporter (SMIT) 1/2 and H^+^-myo-inositol transporter (HMIT) on the cell membrane—both are extensively expressed in the brain and neurons [44]. Berry, G.T. et al. found that SMIT1 is largely responsible for inositol transport during embryonic development [45]. In the present study, several amniotic fluid PIs, particularly PI 35:2, PI 37:4, and PI 40:4, were pronouncedly upregulated after CP versus normal saline treatment (Figure 5 and Figure 6D and Appendix A). This indicates the possibility of impaired embryonic uptake of inositols. However, it needs further validations before a definite conclusion, such as changes in inositol levels in the embryo and evidence for SMIT1 deficit.

### 4.4. TGs

Triglycerides (TGs) are the most abundant lipids and the main energy depot in the body. They are critical for fetal growth during pregnancy. Fetal adipogenesis requires fatty acids synthesized in the liver and those transported from maternal adipose tissue at full term [46]. In early pregnancy, when the fetus is unable to synthesize lipids, essential fatty acids (EFAs), long-chain polyunsaturated fatty acids (LCPUFAs), and other FAs in the maternal circulation are transferred to the fetus across the placenta [47]. Women with elevated TGs during gestation are at higher risk of developing pregnancy complications and adverse perinatal outcomes [48,49]. Increased TG levels are also detectable in small-for-gestational-age (SGA) newborns [50], which can be associated with impaired lipoprotein lipase (LPL) activity and the subsequent development of fetal cellulite. Currently, limited studies have explored amniotic fluid TGs during gestation. Chen, X. et al. [51] analyzed the amniotic fluid lipidomics in pregnant women with hemoglobin Bart’s hydrops fetalis versus pregnant women carrying healthy fetuses. They found that highly unsaturated TGs were significantly lowered, and highly saturated TGs were pronouncedly elevated in those with Hb Bart’s versus healthy controls. In the present study, the metabolism of most amniotic fluid TGs was disturbed after CP treatment versus normal saline, featuring the upregulation of unsaturated TGs, particularly polyunsaturated fatty acids (PUFAs) (three or more double bonds), and the downregulation of saturated TGs (Appendix A and Figure 6C,D). 

PUFAs such as FA 22:4 mentioned above are easily oxidized [52]. This oxidized lipid category, for example, PUFA-containing TGs, can be accumulated, thereby eliciting oxidative stress responses [53]. MDA and SOD are the most frequently used indicators of oxidative stress. MDA is a lipid peroxide resulting from oxygen-free radicals’ attack against lipids. Its content reflects the degree of lipid peroxidation. SOD is a critical antioxidant enzyme that scavenges superoxide anion radicals in the body in order to protect cells from oxygen-free radical damage. In this study, amniotic fluid MDA and SOD levels should have been examined to verify whether CP could trigger maternal–fetal oxidative stress responses. Unfortunately, the volume of available amniotic fluid was too limited to meet the need for the detection system due to high rates of aborted fetuses and stillbirths and low amniotic fluid volume in viable fetuses in the CP group. The placenta is essential for studying the maternal–fetal interface [54]. Therefore, we determined MDA and SOD contents in the placenta to validate CP’s role in maternal–fetal oxidative stress responses. The results showed that MDA content increased and SOD activity decreased significantly after CP treatment. Lipidomics showed that oxidized lipids such as FA 22:4, PC O-38:4, and SM 40:3; 2O were elevated after CP treatment (Appendix A). These results are consistent with the findings of the study conducted by Kim, S.H. et al. [6] that CP induces oxidative stress responses. Oxidative stress has been proven to cause placental dysfunction, resulting in abnormal fetal development [55]. Overall, CP-induced oxidative stress can be a potential mechanism responsible for teratogenic toxicity.

### 4.5. SAs, PAs, and Tryptophan

In GC-MS analysis, we found significantly decreased SA, PA, and tryptophan levels in the amniotic fluid after CP treatment (Figure 7D and Appendix A). SAs are 18-carbon octadecanoic acids, and PAs are 16-carbon hexadecanoic acids. Both of them serve as energy sources that can be rapidly absorbed for embryonic development [56,57]. Keeping the proportion of fatty acids in an appropriate range is vital for embryonic development [58]. It was reported that the exposure of early embryos to high levels of SAs or PAs resulted in metabolic abnormalities [59]. Very low SFA levels are associated with insufficient energy supply and often indicate problems in the development of embryos.

Tryptophan is a neutral amino acid and is one of the essential amino acids [60] for controlling growth, behavior, immune and anti-stress responses, tissue growth, and organ development in men and animals [61,62,63]. Tryptophan and its metabolites have been proven to effectively scavenge free radicals and act as effective antioxidants in the placenta [55]. In this study, amniotic fluid tryptophan was remarkably lowered in pregnant rats receiving CP. This is mainly due to aggravated maternal and fetal oxidative stress by CP, which further disturbs fetal growth and development, as evidenced by lower fetal weight, shorter body length, and a higher stillbirth rate in the CP group.

However, there are also some limitations. Restricted by the nature of nontarget analyses, we utilized relative rather than absolute quantification of omics data. In addition, the representativeness of CP or whether CP’s toxicity can more accurately reflect the actual effects of maternal exposure to other toxicants on the amniotic fluid lipid metabolism during gestation needs to be validated. So do the detailed mechanisms for CP’s toxicity. In our future study, we plan to identify lipids associated with embryotoxicity and oxidative stress-related signalings to explore their roles in cell function and embryonic development.

## 5. Conclusions

Overall, CP exerts teratogenic toxicity on pregnant rats through maternal and fetal oxidative stress. Differentially expressed lipids and metabolites closely associated with CP-induced oxidative stress can be characterized as follows: the accumulation of oxidized lipids in the amniotic fluid, such as VLCUFAs, PUFA-containing TGs, oxidized PCs, and SMs; a lower level of tryptophan, which reduces the efficacy of free radical clearance. CP’s developmental toxicity can be associated with maternal and fetal oxidative stress, as evidenced by an increased MDA concentration and suppressed SOD activity in the placenta. Further, CP-induced lipid imbalance in the amniotic fluid, such as lowered long-chain saturated fatty acids (including stearic acid and palmitic acid) or fewer energy sources, explains the low fetal weight, short body length, and other toxic manifestations. In this study, we utilized a UHPLC-Q-Exactive Orbitrap MS-based approach for lipidomics and metabolomics analysis. This method is worthy of wider application for evaluating the potential toxicity of other agents (toxicants) during embryonic development.

## Figures and Tables

**Figure 1 metabolites-12-01105-f001:**
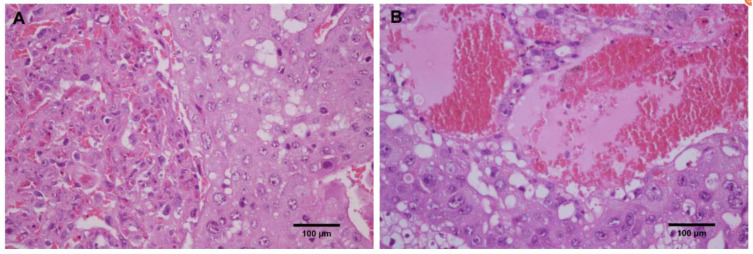
Placental pathology in pregnant rats (magnification, ×200). (**A**) the CK group; (**B**) the CP group.

**Figure 2 metabolites-12-01105-f002:**
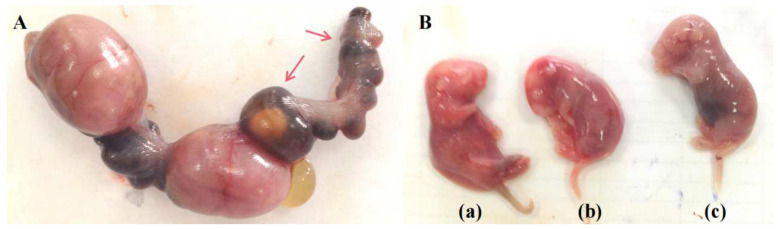
Abnormal embryos and fetuses after CP treatment. (**A**) Fetuses resorbed (red arrows) in the CP group; (**B**) viable fetuses on GD17, (**a**,**b**) two of which are from rats receiving CP, show shorter body length than (**c**) that from a pregnant rat receiving normal saline.

**Figure 3 metabolites-12-01105-f003:**
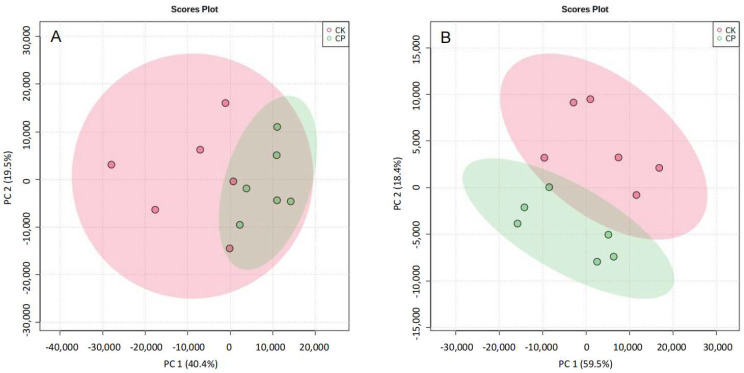
PCA score plots of amniotic fluid lipid profile in the CP and CK groups. Each dot represents an amniotic fluid sample (*n* = 6). (**A**) Positive ion mode; (**B**) negative ion mode.

**Figure 4 metabolites-12-01105-f004:**
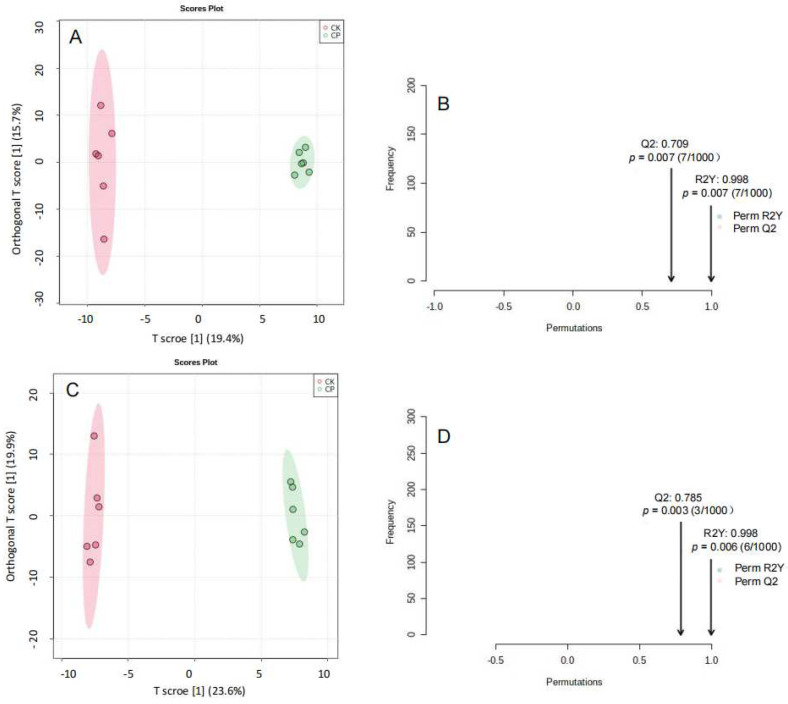
OPLS-DA score plots of amniotic fluid lipid profile and validation of OPLS-DA models in the CP and CK groups using permutation tests (*n* = 6). (**A**,**B**) Positive ion mode; (**C**,**D**) negative ion mode.

**Figure 5 metabolites-12-01105-f005:**
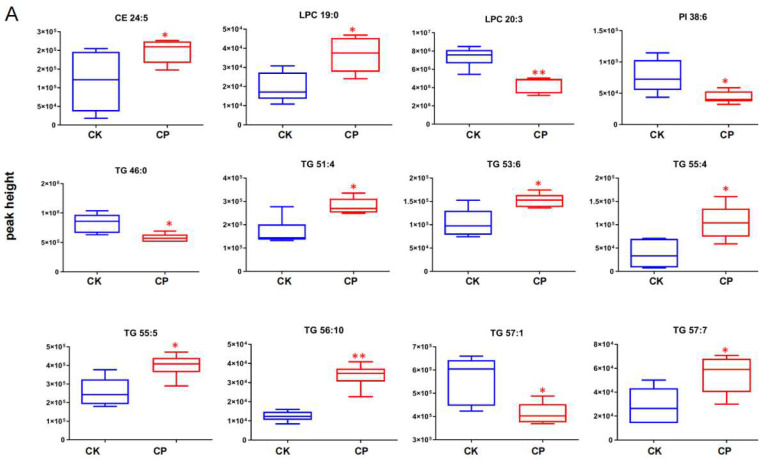
Normalized peak intensity of amniotic fluid lipids after CP treatment (*n* = 6). (**A**) Positive ion mode; (**B**) negative ion mode. *n*= 6. Each data point represents the mean ± S.E.M. * *p* < 0.05 and ** *p* < 0.01, versus CK.

**Figure 6 metabolites-12-01105-f006:**
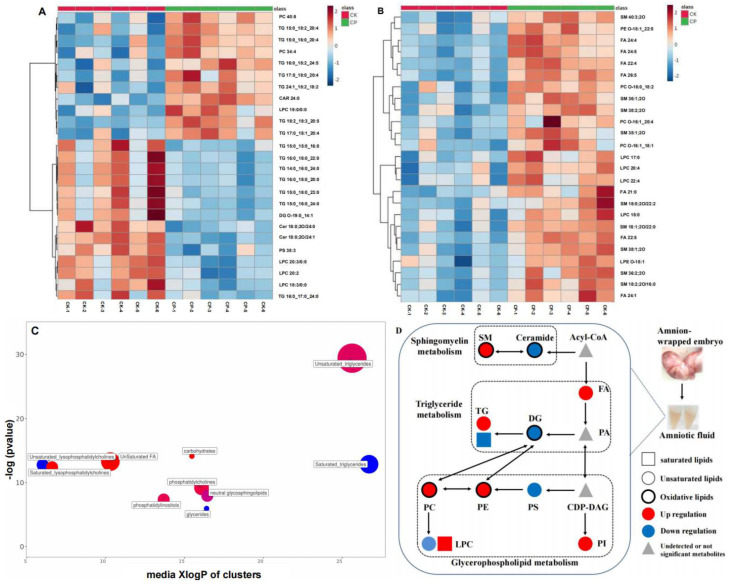
Differentially expressed lipids between the CP and CK groups. Heatmaps of the top 25 differential amniotic fluid lipids in the (**A**) positive and (**B**) negative ion modes (*n* = 6). Each square corresponds to the peak intensity of lipids. Red or blue squares represent increases or decreases in concentration. Dark red or blue squares stand for higher increase or decrease degrees of concentration. (**C**) ChemRICH enrichment analysis of differential lipids (*n* = 6). Each node represents a cluster of differential lipids. Node sizes represent the total number of lipids in each cluster. Red or blue nodes stand for the clusters with most lipids upregulated or downregulated. Deep pink nodes include both upregulated and downregulated lipids. (**D**) Metabolic pathway network of differential lipids. The rectangle represents saturated lipids, the circle represents unsaturated lipids, the thick black border represents oxidized lipids, the blue color represents downregulation, the red color represents upregulation, and the gray triangle represents undetected or not significant lipids.

**Figure 7 metabolites-12-01105-f007:**
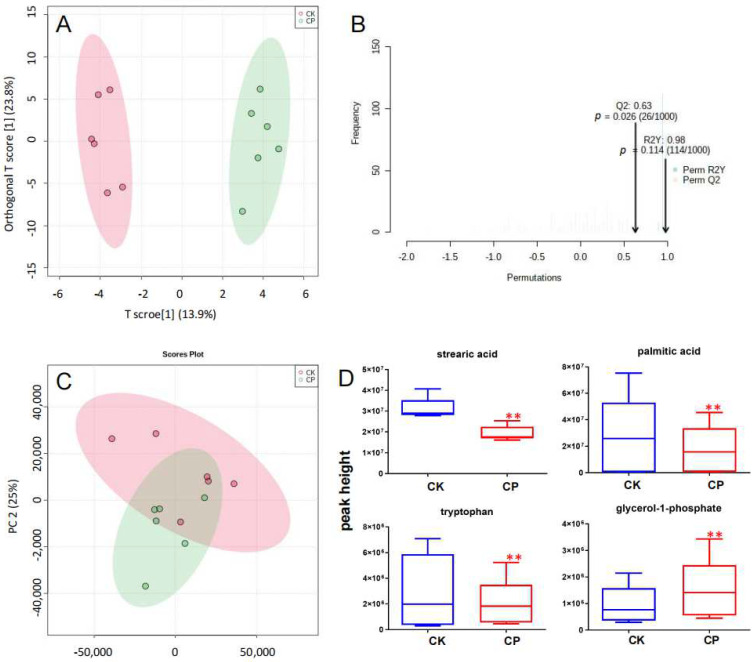
Metabolic profile and differential metabolites of polar substances in the amniotic fluid using GC-MS. (**A**,**B**) OPLS-DA score plots of the amniotic fluid metabolite profile and validation of OPLS-DA models with 1000 permutation tests (*n* = 6). (**C**) PCA score plots of the amniotic fluid metabolite profile (*n* = 6). (**D**) Normalized peak intensity of amniotic fluid metabolites differentially expressed in pregnant rats receiving CP (*n* = 6). Each data point represents the mean ± S.E.M. ** *p* < 0.01 versus CK.

**Figure 8 metabolites-12-01105-f008:**
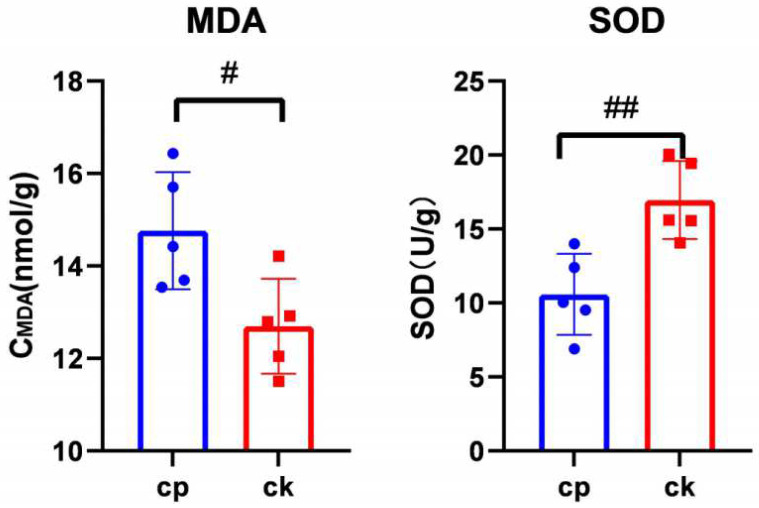
Colorimetric determination of MDA and SOD concentrations in rat placenta (*n* = 5). Each data point represents the mean ± S.E.M. ^#^ *p* < 0.05 and ^##^ *p* < 0.01 versus CK.

**Table 1 metabolites-12-01105-t001:** Effects of CP on fetuses on GD17.

Items	CK	CP 15 mg·kg^−1^
Pregnant rats (N) ^a^	7	9
Maternal weight on GD17 (g; $\bar{x}$ ± *s*)	350.9 ± 25.4	319.3 ± 24.9 *
Fetal and placental weights (g; $\bar{x}$ ± *s*)	27.1 ± 11.3	20.1 ± 13.1 *
Implanted embryos (N) ^a^	87	77
Viable embryos (N) ^a^	80	59
Fetuses resorbed (N [%]) ^b^	6(6.9)	7(9.1)
Late fetal death (N [%]) ^b^	1(1.1)	11(14.3) *
Fetal weight (g; $\bar{x}$ ± *s*)	0.84 ± 0.04	0.55 ± 0.08 **
Body length of fetuses (mm; $\bar{x}$ ± *s*)	20.33 ± 0.46	18.27 ± 1.06 **
Tail length of fetuses (mm; $\bar{x}$ ± *s*)	6.68 ± 0.70	6.56 ± 0.56

^a^ Values are presented as total number; ^b^ values are presented as percent of all implanted embryos. * *p* < 0.05 and ** *p* < 0.01 versus CK.

## Data Availability

Not applicable.

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
