# Peer review of "Cyclophosphamide Induces Lipid and Metabolite Perturbation in Amniotic Fluid during Rat Embryonic Development"

_metabolites, 2022, doi:10.3390/metabo12111105_

Round 1

Reviewer 1 Report

Review comments to the author

Title: ''Cyclophosphamide Induces Lipids and Metabolites Perturbation in Amniotic Fluid During Rat Embryonic Development''.

Manuscript ID: metabolites-1970211.

1. Introduction

1- The author should use recent citations. For example, citation No. [1] can be replaced by the reference (Antioxidants 2019, 8, 415; doi:10.3390/antiox8090415). Please apply this concept as much as possible with the rest of the old citations.

2. Materials and Methods

Chemicals and reagents

1- It is necessary to add a new section containing all the used chemicals and reagents as well as their sources (Company name, city and country).

2.2.2 Chromatographic separation and mass spectrometer settings

1- Page 4, Line 136: The term ''m/z'' should be written in italic font.

Abbreviations:

- List of abbreviations should be inserted by the end of the manuscript before references.

References

1- Page 16, Line 445: Add a dot (.) after the abbreviated word ''Pharm''.

2- Page 16, Line 463:  the abbreviated word ''mol.'' should be written as ''Mol.''.

3- Page 18, Line 533: The author names ''BATTAGLIA, F. C.; MESCHIA, G.; BLECHNER, J. N.; BARRON, D. H.'' should be written as  ''Battaglia, F. C.; Meschia, G.; Blechner, J. N.; Barron, D. H.''.

4- Page 18, Line 546: Add a dot (.) after the abbreviated word ''Res''.

Supplementary material

1- Table S1: The number ''4'' in the formula ''NH4'' should be written in subscript font.

2- Table S1: The number ''3'' in the formula ''CH3COO'' should be written in subscript font.

Author Response

Thanks very much for your comments. They are all valuable and very helpful for revising and improving our manuscript. We have carefully revised the manuscript strictly based on all the comments and suggestions point by point which we hope meet with approval.

Reviewer 2 Report

Xu et al present a manuscript on the metabolomic/lipidomic analysis of rat amniotic fluid for rats exposed to cyclophosphamide. The paper is well written and easy to follow. I believe it would be of interest to the general readership of metabolites and wide. I have some minor comments for the authors to address detailed below:

Line 110: Add the word liquid before chromatography is using LC-MS acronym.

Line 116: “300 300 r.min-1” deleted the repeated 300

Line 130: Was this the polar fraction? Also include the methods in full and keep the reference.

Table1: Are these mean and standard deviation? I would add this to the caption to help the reader.

Figure 1: this needs a scale bar adding.

Figure 4B, 4D and 7D the colours chosen do not display in the figure.

Line 222: Change to MS2 rather than secondary ion MS spectrometry which is a different technique.

Figure S1: There seems to be high background here why is that?

Figure S5: There seems to be instrumental drift with the QC, is this because of source fouling?

Table S1: Can the authors add a PPM mass error for the metabolites.

Author Response

(The authors gave the same response as above.)

Reviewer 3 Report

In this work, the authors mainly performed lipidomics and metabolomics studies on the amniotic fluid lipids from rat treated with or without cyclophosphamide (CP). It was found that CP induced the perturbation of some lipids and metabolite, mainly including the accumulation of oxidized lipids in the amniotic fluid, such as VLCUFAs, PUFA-containing TGs, oxidized PCs and SMs, and a lower level of tryptophan.

After carefully reading the manuscript, the overall feeling is that this is an "assembly line" work, lacking of thinking and limited innovation. In addition, the main defects of this work are as follows:

1. This work is extremely lacking in structural identification of disordered lipids. As a mass spectrometry work, the lack of MS and MS/MS data and interpretation of disordered lipids greatly reduces the credibility of the data in this work.

2. This work has found so many changes in lipids, could there be a relatively systematic network to explain the causes and consequences of these changes in lipids?

Author Response

(The authors gave the same response as above.)

Round 2

Reviewer 3 Report

All questions raised were responded.  No more questions.